# Endothelial activation and stress index in risk stratification and treatment optimization for critically ill patients with acute kidney injury: A retrospective cohort study from MIMIC-IV database

**Chengshu Liang[1], Kai Wang[2]\***

**1** Department of Nephrology, Wuxi County People's Hospital, Chongqing, China, **2** Department of Cardiology, The Second Affiliated Hospital of Chongqing Medical University, Chongqing, China

\* nkuwangkai@163.com

## Abstract

### Background

Endothelial dysfunction is critical in the pathogenesis of acute kidney injury (AKI). It sought to investigate the role of endothelial activation and stress index (EASIX) in risk stratification and treatment optimization for critically ill patients with AKI.

### Methods

Utilizing MIMIC-IV 3.1 database, a retrospective cohort study was undertaken. Given the non-normal distribution, EASIX was transformed logarithmically. The endpoints were 1-year and ICU all-cause mortality. The association was assessed using Kaplan-Meier curves, Cox models, restricted cubic splines and propensity score via overlap weights. Subgroup analyses were conducted to assess optimal population for EASIX application and to preliminarily explore its potential role in guiding treatment strategy optimization.

### Results

It comprised 17624 patients with AKI, exhibiting 1-year and ICU mortality rates of 44.6% and 19.2%. Elevated log2(EASIX) levels were independently associated with an increased 1-year mortality (HR: 1.41, 95% CI: 1.32–1.50) and ICU mortality (RR: 1.49, 95% CI: 1.38–1.62), as a finding corroborated by overlap-weighted propensity score analysis. Subgroup analyses indicated a stronger association in patients without severe AKI, CKD, sepsis or CRRT, and patients with lower levels of age or Acute Physiology Score (APS) III and higher levels of albumin ($p < 0.05$ for all). The glucocorticoid use may be independently associated with an increased risk of 1-year (HR: 1.27, 95% CI: 1.21–1.34) and ICU (HR: 1.39, 95% CI: 1.31–1.47) mortality.

**Data availability statement:** The datasets presented can be obtained from the following link on the premise of completing MIMIC database's training (https://doi.org/10.6084/m9.figshare.31981221).

**Funding:** The author(s) received no specific funding for this work.

**Competing interests:** The authors declare no competing interests.

The glucocorticoid-associated risk decreased as the log2(EASIX) level increased ($p < 0.001$).

## Conclusion

It found the positive association between log2(EASIX) levels and risk of mortality in critically ill patients suffering from AKI, particularly in those with decreased age or APS III, elevated albumin, and those characterized by mild AKI, or absence of CKD, sepsis or CRRT. These findings underscored the significance of EASIX in enhancing risk stratification systems and in guiding personalized anti-inflammatory treatment strategies.

## Introduction

Acute kidney injury (AKI), as patient's critical complication in intensive care unit (ICU), exacerbates the severity of the primary disease and contributes independently to mortality [1]. It is particularly evident in patients with underlying medical conditions such as sepsis and chronic kidney disease (CKD) [2]. Research indicated that the restoration of kidney function in AKI patients was suboptimal, with approximately 25% of survivors developing acute kidney disease, while 33% progressing to CKD in five years and often necessitating long-term renal replacement therapy [3,4]. Despite advances in critical care technology, the management of AKI remains challenging, and there are no specific measures to effectively reverse it. Management strategies for AKI focus on early detection and intervention to prevent kidney damage from progressing to more severe stages. Therefore, early identification and management of risk factors, as well as the search for possible directions for intervention, are essential to improve the prognosis of patients with AKI in ICU setting.

Recent studies have highlighted that endothelial dysfunction may be a key factor in the pathogenesis of AKI [5]. Specifically, endothelial damage may lead to increased vascular permeability of the renal microcirculation, increased inflammatory response, microvascular dysfunction, and subsequent tissue damage [6]. Endothelial dysfunction also plays a key role in distant organ dysfunction during AKI, including endothelial damage, production of reactive oxygen species, and increased inflammatory response [7]. Endothelial dysfunction and injury can increase the secretion of lactate dehydrogenase (LDH), decrease the platelet count due to the coagulation cascade activation [8], and affect the glomerular endothelium leading to renal failure and elevated creatinine levels [9]. Comprising serum creatinine, platelet and LDH levels, the endothelial activation and stress index (EASIX) was originally created to gauge the severity of endothelial pathology after stem cell transplants [10]. Researches have shown that higher EASIX levels were linked to a greater risk of mortality in patients receiving allogeneic stem cell transplants, a situation that was related to thrombotic microangiopathy caused by endothelial dysfunction [11,12]. Additional research has identified EASIX as a promising marker for individuals suffering from severe liver disease [13], heart failure [14], sepsis [15], and traumatic brain injury

[16]. Further investigations have demonstrated an association between high EASIX levels and negative results in patients suffering from coronary artery disease (CAD) [17], particularly myocardial infarction [18]. Notably, recent study has confirmed EASIX as an independent risk factor for new-onset AKI in critically ill cancer patients [19]. However, the connection between EASIX and the outcomes of critically ill patients with AKI has yet to be investigated. Importantly, how EASIX may be able to guide the optimization of treatment strategies remains unexplored.

To fill these gaps, we ran an extensive analysis utilizing the MIMIC-IV database, including subgroup analyses to identify optimal population for EASIX application and to initially investigate how EASIX may guide treatment strategy selection.

## Materials and methods

### Study population

The population was derived from MIMIC-IV database (version 3.1) from 2008 to 2022. The corresponding author has certified access to this database (ID: 64734176). The study involving human participants adhered to the ethical guidelines set by institutional and national research committees, along with the Helsinki Declaration. Approval for the study was granted by the Human Institutional Review Board at Beth Israel Deaconess Medical Center. The further ethical review was considered unnecessary for this study, as no additional data collection was undertaken. The individual patient consent was deemed unnecessary for this study, as the study was retrospective and the anonymized data was publicly available, and individuals could not be identified either directly or indirectly.

Patients with AKI as a first-time ICU admission were enrolled. Specifically, AKI was diagnosed according to the KDIGO guidelines based on the level of serum creatinine and urine output after ICU admission, but not according to the International Classification of Diseases [20,21]. Exclusions were made for those under 18, ICU stays under 24 hours, or missing data on serum creatinine, serum LDH, or platelets (Fig 1).

### Data collection

Clinical data were initially collected for the first measurement in 24 hours of ICU admission. Variables with over 20% missing data were removed. To record age, gender, blood pressure, saturation of peripheral oxygen (SpO2), respiratory and heart rate, comorbidities (hypertension, heart failure, diabetes, CKD, cirrhosis, sepsis, cancer), and laboratory markers (platelets, hemoglobin, RDW, white blood cell count (WBC), aspartate aminotransferase (AST), alanine aminotransferase (ALT), total bilirubin (TBil), albumin, total protein, creatinine, blood urea nitrogen (BUN), LDH, lactate, anion gap, serum electrolytes and arterial blood gases), stage of AKI, disease severity scores (SOFA and APS III), as well as medication details (renin angiotensin aldosterone system inhibitors (RAASi), glucocorticoid, antibiotic), ventilation and continuous renal replacement therapy (CRRT).

Exposure was EASIX derived from the formula, LDH (U/L) × serum creatinine (mg/dL) / platelets (10^9/L) [22]. The primary endpoint was all-cause mortality within one year, and the second endpoint was all-cause ICU mortality.

### Statistical analysis

The multiple imputation approach was utilized for overcoming the missing data. Continuous features were represented by median with quartiles or mean with standard deviation, when categorical features were expressed as count with percentage. Given the non-normal distribution, EASIX was transformed logarithmically (into log2(EASIX)) [14]. All analyses were performed using R (version 4.4.1), and significance threshold was set at $p < 0.05$.

To begin with, by utilizing the maximally selected rank statistics in the 'survminer' package, the optimal threshold for log2(EASIX) was set to alert on mortality risk. The assessment of mortality risk (hazard ratio (HR)) was conducted using Kaplan-Meier curves, alongside Cox models minimizing confounding bias. Introduction of covariates was guided by previous research or insights from clinical practice, rather than relying solely on data-driven approaches [23]. These included

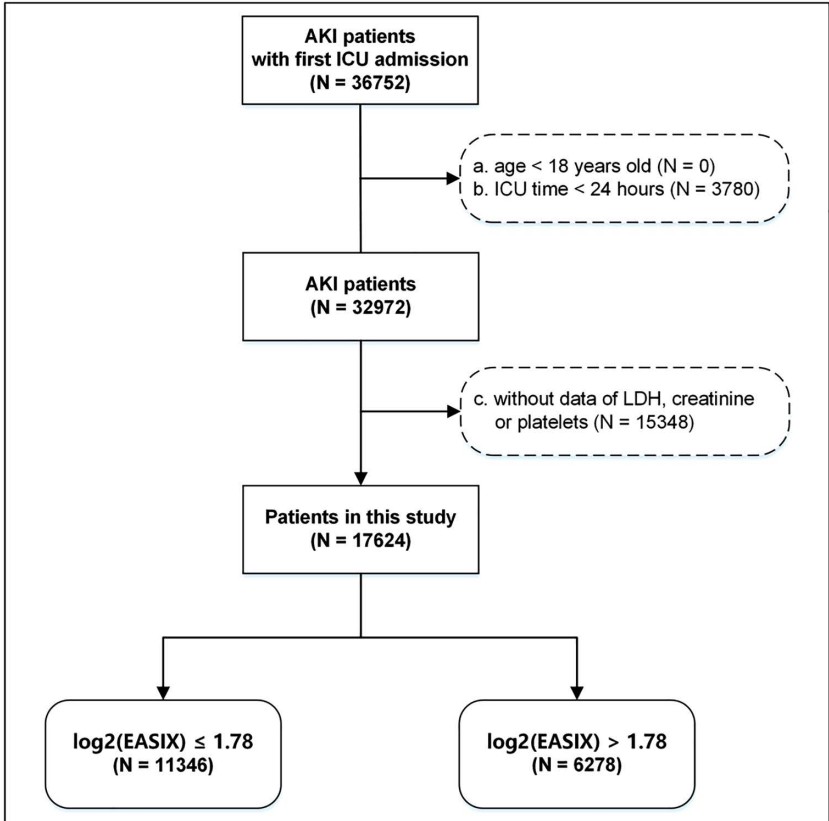

**Fig 1. Flow chart of study population inclusion.**

demographic factors such as age and gender; comorbidities including hypertension, diabetes, CKD, heart failure, cirrhosis, sepsis, and cancer; indicators of anemia and inflammation, such as hemoglobin, platelets, WBC, RDW; as well as liver and kidney function parameters, namely AST, ALT, TBil, albumin, BUN, creatinine, and LDH. Additionally, lactate and anion gap were assessed for internal environment evaluation; therapeutic factors included glucocorticoids, RAASi, CRRT, and ventilation; and severity of the disease was characterized by SOFA and APS III. To detect multicollinearity, variance inflation factors were computed, with values exceeding 5 indicating a possibility of multicollinearity among these variables. Modified Poisson regression was used to calculate the relative risk (RR) for log2(EASIX), considering both ICU mortality and 1-year mortality as outcomes to ensure the analysis reliability.

Then, subgroup analysis was conducted to calculate the HR across various patients. For the three continuous variables, age, APS III, or albumin, interactive RCS were employed to assess the continuous interaction between them an log2(EASIX) [24]. Furthermore, considering that endothelial impairment reflected in EASIX is strongly associated with inflammation and oxidative stress, and anti-inflammatory therapy may confer benefits, interactive RCS was utilized to evaluate the moderating effect of log2(EASIX) on the treatment effect of glucocorticoid within this population. These were conducted using the 'interactionRCS' package.

Afterwards, the Boruta algorithm was used to evaluate and compare the contribution of log2 (EASIX) and other baseline features to the prediction for two outcome event. The Boruta algorithm determines the crucial features by evaluating Z-values of features against Z-values of a "shadow feature," which is produced by randomly scrambling the real feature. The real-world feature is labeled as "important" when its Z-value exceeds the highest Z-value of the shadow feature

in multiple independent tests. Alternatively, it is identified as "unimportant." These were employed using the 'Boruta' package.

Finally, for the robustness of analysis, all baseline information except LDH, creatinine and platelets was controlled using overlap-weighting probability score. These three exclusions were due to the interdependence among them and log2(EASIX). Overlap-weighting Kaplan-Meier survival curves were used to compare the effect of log2(EASIX) on risk of all-cause mortality. These analyses were conducted using the 'WeightIt' package.

## Results

### Patient characteristics

This study involved 17624 individuals from the MIMIC-IV (Fig 1). A log2(EASIX) cut-off point of 1.78 was determined as optimal for evaluating the risk of all-cause mortality (Fig 2A).

In this population, patients with elevated log2(EASIX) demonstrated increased levels of AKI stage, RDW, kidney and liver injury, lactate, anion gap, LDH, APS III and SOFA, higher prevalence of conditions such as diabetes, cirrhosis, sepsis, heart failure, CKD, use of glucocorticoid, antibiotic or CRRT (Table 1). The mortality rates in the ICU and after one year were 19.2% (3375/17624) and 44.6% (7861/17624). Patients with elevated log2(EASIX) exhibited increased ICU mortality (29.7%) and 1-year mortality (57.2%) ($p < 0.001$ for both comparisons).

### Kaplan-Meier survival curve

Based on a cut-off value of 1.78 of log2(EASIX), Kaplan-Meier analysis was utilized. Those exhibiting higher log2(EASIX) levels were found to have a significantly poorer prognosis (Log-rank test $p < 0.001$) (Fig 2B).

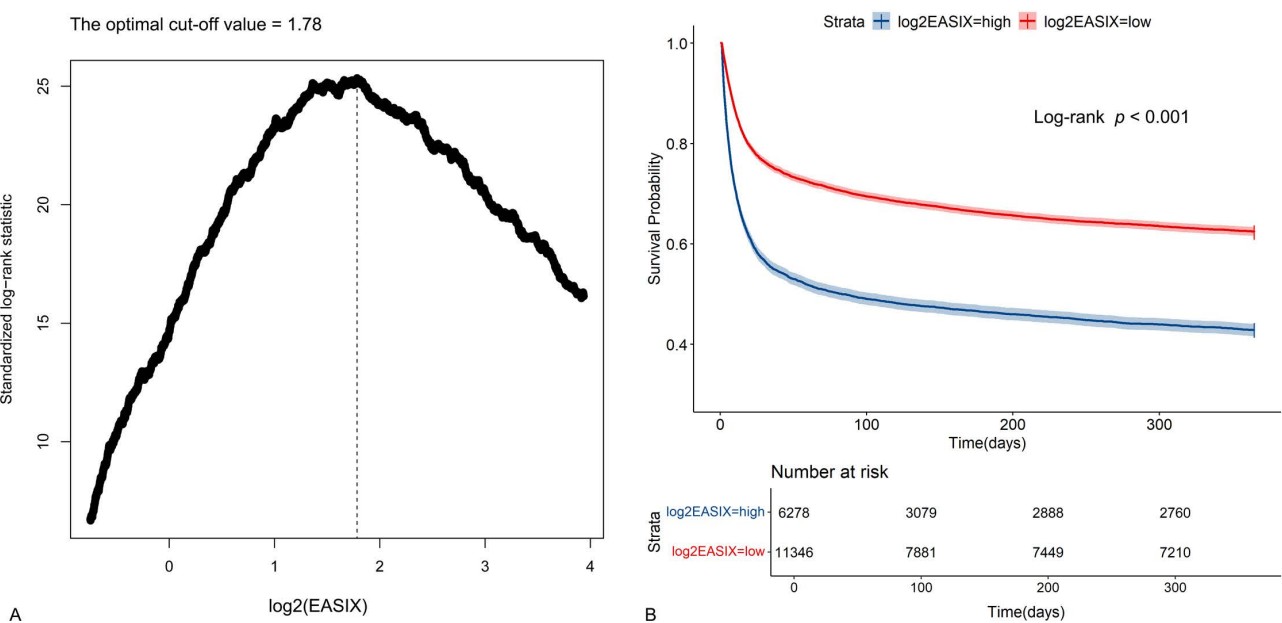

**Fig 2. Identification of optimal cutoff value of log2(EASIX) and plotting for the corresponding Kaplan-Meier curves. (A)** The optimal cutoff value of log2(EASIX) as an warning indicator of mortality risk was determined using the maximally selected rank statistics. **(B)** Kaplan-Meier curves estimated the probability of survival by the levels of log2(EASIX).

**Table 1. Baseline information according to the levels of log2(EASIX).**

| Characteristics | Overall, n = 17624 | Low (≤ 1.78), n = 11346 | High (> 1.78), n = 6278 | p value |
|---|---|---|---|---|
| age (years) | 66.41 ± 16.03 | 66.99 ± 16.13 | 65.36 ± 15.79 | <0.001 |
| female | 7390 (41.9%) | 5060 (44.6%) | 2330 (37.1%) | <0.001 |
| respiratory rate (/min) | 20.10 ± 6.58 | 19.88 ± 6.50 | 20.51 ± 6.72 | <0.001 |
| heart rate (/min) | 91.32 ± 21.30 | 90.81 ± 21.12 | 92.25 ± 21.60 | <0.001 |
| systolic blood pressure (mmHg) | 120.96 ± 25.34 | 122.15 ± 24.94 | 118.81 ± 25.92 | <0.001 |
| diastolic blood pressure (mmHg) | 67.00 (56.00, 79.00) | 68.00 (57.00, 80.00) | 65.00 (54.00, 78.00) | <0.001 |
| saturation of peripheral oxygen (%) | 98.00 (95.00, 100.00) | 98.00 (95.00, 100.00) | 97.00 (95.00, 100.00) | <0.001 |
| hypertension | 6695 (38.0%) | 5012 (44.2%) | 1683 (26.8%) | <0.001 |
| diabetes | 5740 (32.6%) | 3457 (30.5%) | 2283 (36.4%) | <0.001 |
| chronic kidney disease | 4013 (22.8%) | 1857 (16.4%) | 2156 (34.3%) | <0.001 |
| AKI stage | | | | <0.001 |
| 1 | 3864 (21.9%) | 2630 (23.2%) | 1234 (19.7%) | |
| 2 | 7527 (42.7%) | 5741 (50.6%) | 1786 (28.4%) | |
| 3 | 6233 (35.4%) | 2975 (26.2%) | 3258 (51.9%) | |
| heart failure | 5929 (33.6%) | 3522 (31.0%) | 2407 (38.3%) | <0.001 |
| cirrhosis | 2092 (11.9%) | 887 (7.8%) | 1205 (19.2%) | <0.001 |
| sepsis | 11688 (66.3%) | 7050 (62.1%) | 4638 (73.9%) | <0.001 |
| cancer | 2939 (16.7%) | 1961 (17.3%) | 978 (15.6%) | 0.004 |
| white blood cell (10^9/L) | 11.30 (7.90, 16.20) | 11.40 (8.10, 16.00) | 11.10 (7.20, 16.70) | <0.001 |
| hemoglobin (g/dL) | 10.45 ± 2.38 | 10.72 ± 2.30 | 9.96 ± 2.44 | <0.001 |
| red cell distribution width (%) | 14.90 (13.70, 16.80) | 14.60 (13.50, 16.10) | 15.70 (14.30, 17.70) | <0.001 |
| platelets (10^9/L) | 182.00 (123.00, 255.00) | 211.00 (155.00, 284.00) | 127.00 (76.00, 189.75) | <0.001 |
| alanine aminotransferase (U/L) | 27.00 (16.00, 62.00) | 23.00 (14.00, 45.00) | 41.00 (20.00, 131.00) | <0.001 |
| aspartate aminotransferase (U/L) | 41.00 (24.00, 94.00) | 33.00 (21.00, 60.00) | 76.00 (35.00, 247.00) | <0.001 |
| albumin (g/dL) | 3.01 ± 0.64 | 3.05 ± 0.64 | 2.94 ± 0.63 | <0.001 |
| total bilirubin (mg/dL) | 0.70 (0.40, 1.40) | 0.60 (0.40, 1.00) | 0.90 (0.50, 2.40) | <0.001 |
| creatinine (mg/dL) | 1.10 (0.80, 1.90) | 0.90 (0.70, 1.30) | 2.10 (1.30, 3.80) | <0.001 |
| blood urea nitrogen (mg/dL) | 23.00 (15.00, 40.00) | 19.00 (13.00, 28.00) | 39.00 (24.00, 62.00) | <0.001 |
| lactate dehydrogenase (U/L) | 288.00 (214.00, 430.00) | 252.00 (197.00, 331.00) | 429.00 (286.00, 773.75) | <0.001 |
| anion gap (mmol/L) | 15.17 ± 4.88 | 13.93 ± 3.93 | 17.42 ± 5.58 | <0.001 |
| lactate (mmol/L) | 1.80 (1.20, 2.80) | 1.60 (1.20, 2.50) | 2.20 (1.40, 3.70) | <0.001 |
| SOFA | 6.14 ± 3.80 | 4.87 ± 3.11 | 8.44 ± 3.85 | <0.001 |
| APS III | 52.68 ± 22.39 | 46.74 ± 19.43 | 63.41 ± 23.38 | <0.001 |
| glucocorticoid | 4729 (26.8%) | 2711 (23.9%) | 2018 (32.1%) | <0.001 |
| antibiotic | 15192 (86.2%) | 9593 (84.5%) | 5599 (89.2%) | <0.001 |
| inhibitors of renin angiotensin aldosterone system | 5152 (29.2%) | 3581 (31.6%) | 1571 (25.0%) | <0.001 |
| continuous renal replacement therapy | 1651 (9.4%) | 352 (3.1%) | 1299 (20.7%) | <0.001 |
| ventilation | 14834 (84.2%) | 9604 (84.6%) | 5230 (83.3%) | 0.020 |
| ICU mortality | 3375 (19.2%) | 1511 (13.3%) | 1864 (29.7%) | <0.001 |
| 1-year mortality | 7861 (44.6%) | 4271 (37.6%) | 3590 (57.2%) | <0.001 |

## Risk ratios for all-cause mortality

After controlling for potential confounding factors, all variance inflation factors remained below 5, suggesting that multicollinearity was not an issue (S1 Fig). In model 3, elevated log2(EASIX) were linked to an increased mortality (HR: 1.41, 95% CI: 1.32–1.50), and a unit increase in log2(EASIX) was linked to a 17% higher risk. Consistent findings were observed across models 0, 1, and 2 (Table 2).

The modified Poisson regression found that elevated log2(EASIX) levels were analogously connected to a higher risk of ICU mortality (RR: 1.49, 95% CI: 1.38–1.62), and of 1-year mortality (RR: 1.22, 95% CI: 1.17–1.27) (Table 2). Models 0, 1, and 2 showed consistent patterns.

Furthermore, an increase in log2(EASIX) levels was associated with a corresponding and linear rise in all-cause mortality risk, highlighting the reliability of log2(EASIX) as a significant marker in this demographic ($p$ for nonlinear = 0.255) (Fig 3A).

## Subgroup analysis

The subgroup analysis assessed the link between log2(EASIX) levels and the risk of 1-year mortality across different cohorts. This positive association was uniformly observed across various subgroups categorized by gender, diabetes status, hypertension, heart failure, and ventilation conditions (Fig 3B). Conversely, the link was notably stronger in those experiencing mild AKI ($p$ for interaction = 0.007), those who did not have CKD ($p$ = 0.047), CRRT ($p$ = 0.016), sepsis ($p$ < 0.001) or glucocorticoid use ($p$ < 0.001). Additionally, younger patients ($p$ = 0.001) and those classified with lower APS III ($p$ = 0.005), as well as individuals with higher albumin levels ($p$ = 0.011), exhibited a more pronounced association (Fig 4A-C).

About a quarter of patients (26.8%) of patients underwent glucocorticoid therapy. After minimizing confounding bias, this glucocorticoid use was independently associated with an increased risk of 1-year (HR: 1.27, 95% CI: 1.21–1.34) and

**Table 2. The association between log2(EASIX) and all-cause mortality.**

| log2(EASIX) | Number | Model 0 | Model 1 | Model 2 | Model 3 |
|---|---|---|---|---|---|
| cut-off value = 1.78 | | **Hazard Ratio for 1-year mortality (Cox regression)** | | | |
| Low (≤ 1.78) | 11346 | reference | reference | reference | reference |
| High (>1.78) | 6278 | 1.89 (1.81,1.97) | 1.65 (1.57,1.73) | 1.48 (1.39,1.57) | 1.41 (1.32,1.50) |
| Each unit increase | 17624 | 1.17 (1.16,1.18) | 1.14 (1.13,1.16) | 1.20 (1.17,1.23) | 1.17 (1.15,1.20) |
| cut-off value = 1.78 | | **Relative Risk for ICU mortality (modified Poisson regression)** | | | |
| Low (≤ 1.78) | 11346 | reference | reference | reference | reference |
| High (>1.78) | 6278 | 2.23 (2.10,2.37) | 1.76 (1.65,1.88) | 1.61 (1.48,1.74) | 1.49 (1.38,1.62) |
| Each unit increase | 17624 | 1.22 (1.20,1.23) | 1.15 (1.14,1.17) | 1.23 (1.20,1.26) | 1.19 (1.15,1.22) |
| cut-off value = 1.78 | | **Relative Risk for 1-year mortality (modified Poisson regression)** | | | |
| Low (≤ 1.78) | 11346 | reference | reference | reference | reference |
| High (>1.78) | 6278 | 1.52 (1.47,1.57) | 1.36 (1.31,1.40) | 1.25 (1.20,1.31) | 1.22 (1.17,1.27) |
| Each unit increase | 17624 | 1.10 (1.10,1.11) | 1.08 (1.07,1.09) | 1.10 (1.08,1.12) | 1.09 (1.07,1.11) |

model 0: only log2(EASIX) was included.

model 1: age, gender, diabetes, hypertension, AKI stage, CKD, heart failure, cirrhosis, sepsis, and cancer were adjusted.

model 2: age, gender, diabetes, hypertension, AKI stage, CKD, heart failure, cirrhosis, sepsis, and cancer, as well as WBC, hemoglobin, RDW, platelets, AST, ALT, TBil, albumin, creatinine, BUN, LDH, lactate, anion gap, glucocorticoid, antibiotic, RAASi, ventilation, and CRRT were adjusted.

model 3: age, gender, diabetes, hypertension, AKI stage, CKD, heart failure, cirrhosis, sepsis, cancer, WBC, hemoglobin, RDW, platelets, AST, ALT, TBil, albumin, creatinine, BUN, LDH, lactate, anion gap, antibiotic, RAASi, ventilation, and CRRT, as well as SOFA and APS III were adjusted.

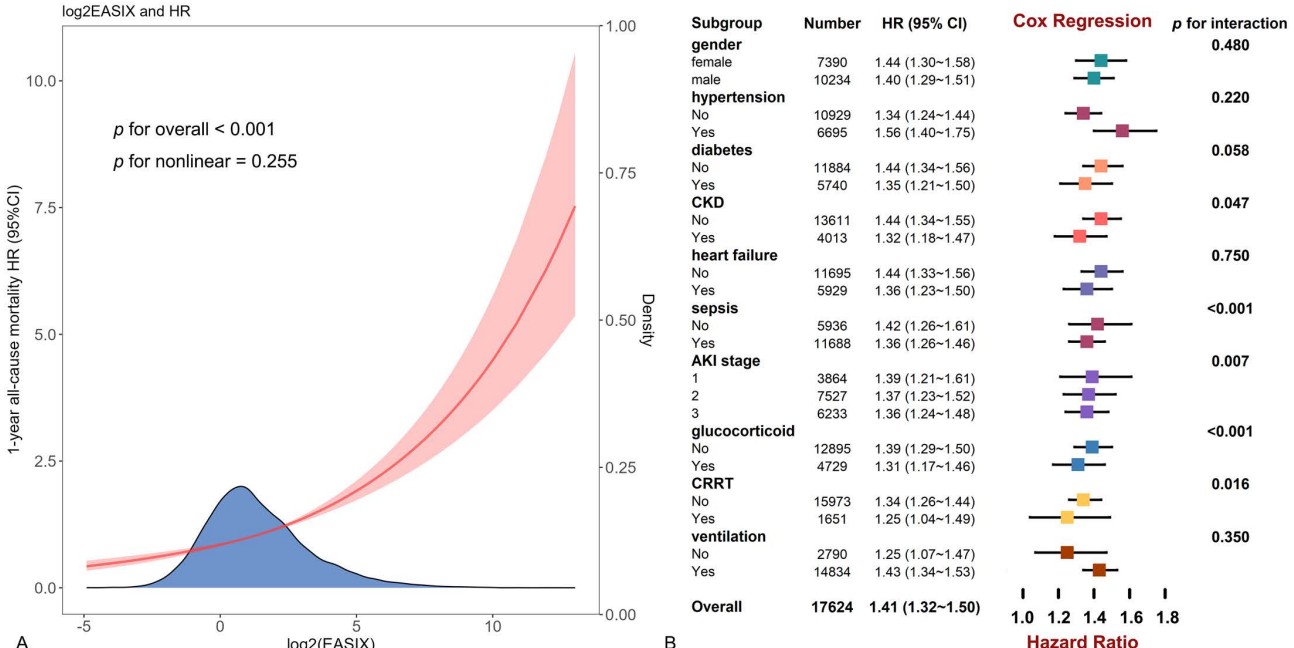

**Fig 3. The restricted cubic spline curve for all patients and forest plot for subgroup analysis. (A)** Restricted cubic spline curve of the log2(EASIX) and hazard ratios in patients with AKI. **(B)** Subgroup analysis by gender, diabetes, hypertension, AKI stage, heart failure, sepsis, chronic kidney disease, glucocorticoid use, continuous renal replacement therapy, and ventilation.

ICU (HR: 1.39, 95% CI: 1.31–1.47) mortality (S1 Table). Meanwhile, the glucocorticoid-associated risk decreased as the log2(EASIX) level increased (*p* for interaction < 0.001) (Fig 4D, S1 Table).

## Boruta algorithm

Using the Boruta algorithm for the evaluation of the contribution to predicting the risk of mortality at 1 year, log2 (EASIX) ranked eighth among the all factors and fourth among all laboratory test factors (S2 Fig). Meanwhile, log2 (EASIX) ranked sixth among all characteristics and second among all laboratory indicators in the evaluation of the contribution to predicting the risk of ICU mortality (S2 Fig). These implied that log2 (EASIX) may serve as critical role in adverse outcomes prediction in critically ill patients with AKI.

## Sensitivity analysis

The overlap-weighting probability score was utilized to establish equilibrium across all baseline variables, with the exceptions of LDH, creatinine, and platelets (Fig 5A). Following this, Kaplan-Meier curves revealed outcomes that aligned with the prior section, suggesting that patients presenting high log2(EASIX) had an elevated risk of mortality (adjusted HR: 1.22, 95% CI: 1.15–1.29, *p* < 0.001) (Fig 5B).

## Discussion

This study undertook a thorough examination of the positive link between log2(EASIX) levels and risk of mortality in critically ill patients diagnosed with AKI. It identified a more significant impact in those who were relatively younger or had lower APS III, higher albumin levels, and those categorized with mild AKI, without CKD, sepsis, or CRRT. Notably, this association may inform the optimization of anti-inflammatory treatment strategies for these patients.

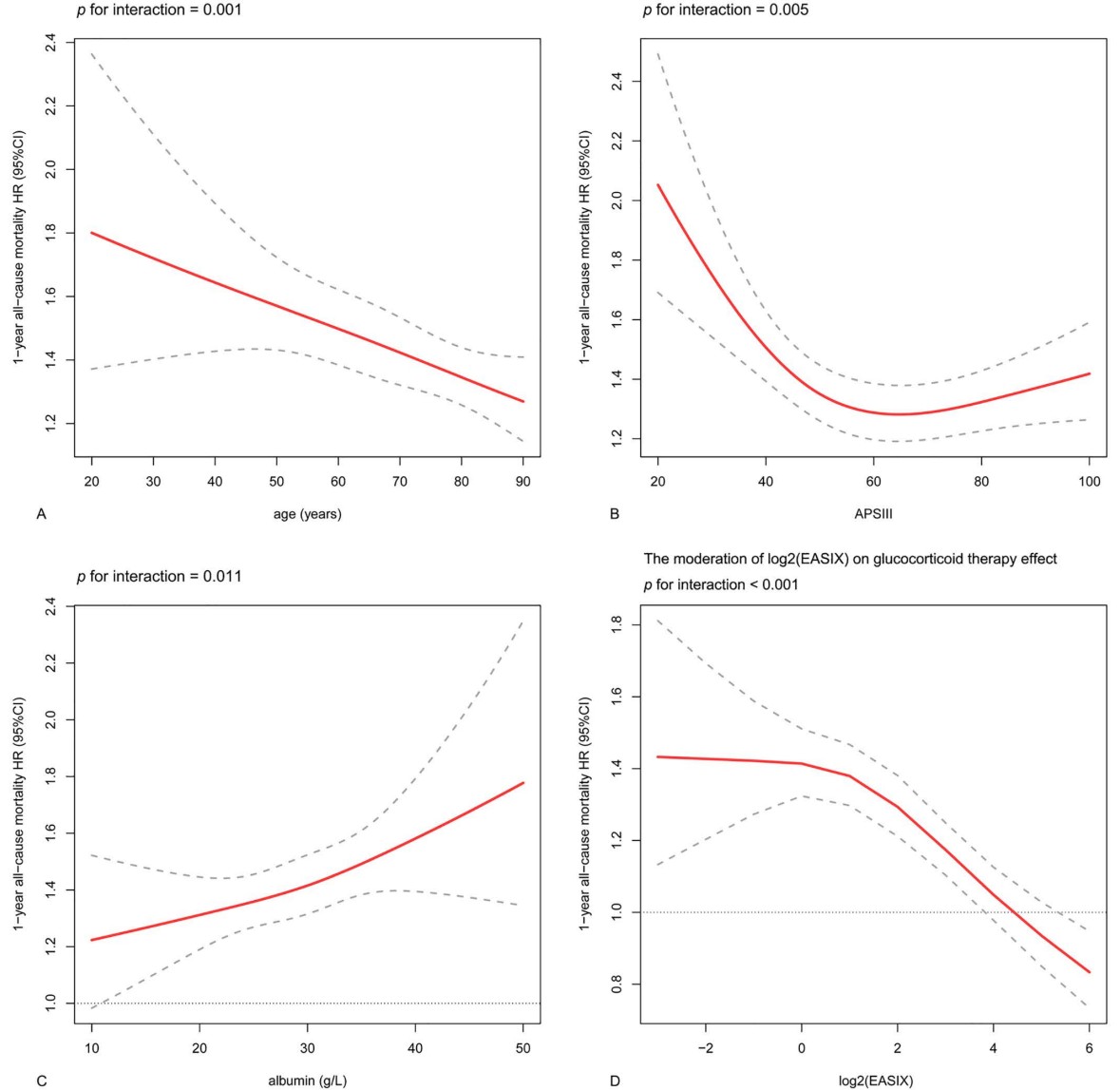

**Fig 4. The interactive restricted cubic spline curves for subgroup analysis. (A)** The analysis in various levels of age. **(B)** The analysis in various levels of APS III**. (C)** The analysis in various levels of albumin. **(D)** The elevated log2(EASIX) levels mitigated the adverse effects associated with glucocorticoid therapy.

EASIX, offering a straightforward yet effective measure of endothelial cell activation and damage, has been extensively investigated across various clinical contexts. One of the primary applications of EASIX is in predicting mortality and complications in patients undergoing allogeneic hematopoietic stem cell transplantation, and research indicated that elevated EASIX scores were associated with increased mortality [11]. Additionally, EASIX has been validated as a prognostic indicator in patients with CAD, effectively predicting mortality regardless of whether it was measured before or after coronary catheterization [17]. In the context of acute pancreatitis, EASIX has demonstrated a correlation with all-cause mortality, with its predictive performance comparable to other established scoring systems, underscoring its potential as a prognostic marker in intensive care settings [22]. These findings underscore the importance of EASIX for assessing

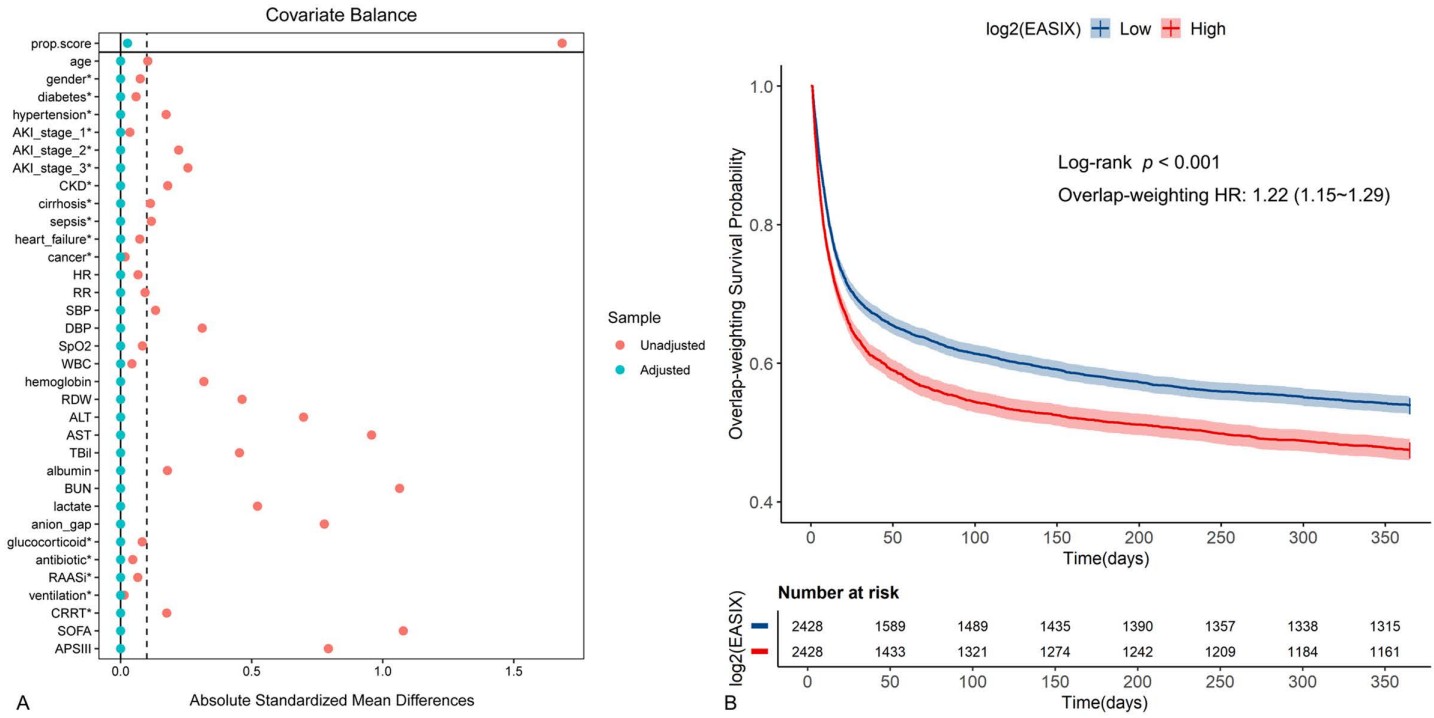

**Fig 5. The covariate-adjusted Kaplan-Meier curves estimated the probability of survival by the levels of log2(EASIX). (A)** Under the overlap weighting, all relevant variables were balanced between the two groups. **(B)** Kaplan-Meier curves were adjusted for all the baseline covariates in Table 1, except LDH, creatinine and platelets due to the interdependence among them and log2(EASIX).

endothelial injury and stress levels. Consistently, we looked at the relationship between EASIX and the mortality risk in ICU patients with AKI. It is noteworthy that AKI is sometimes underdiagnosed at discharge in the MIMIC database from the International Classification of Diseases. Consequently, the diagnosis and staging of AKI were determined according to the KDIGO guidelines, based on the patient's serum creatinine or urine output during hospitalization. Thorough evaluations consistently indicated that individuals exhibiting increased log2(EASIX) encountered a higher likelihood of mortality. This connection remained unaffected by compromised liver and kidney function, as well as the severity of the disease. Moreover, the analysis results of the Boruta algorithm support that log2(EASIX) takes on irreplaceable tasks in the evaluation of adverse event risk in these patients. Sensitivity analysis supported these findings. In the future, the inclusion of a variety of integrated and composite indicators, such as log2(EASIX) and blood urea nitrogen to albumin ratio [25], into the prediction model may bring about a leapfrog improvement in risk early warning capabilities. Consequently, we recommend utilizing EASIX for evaluating endothelial impairment and the risk stratification in critically ill patients with AKI.

Our observations indicated that the prognostic impact of EASIX was particularly pronounced in AKI patients characterized by younger age or decreased APS III, elevated albumin, mild AKI, and absence of CKD, sepsis or CRRT. Although APS III highlights the evaluation of damage to vital organs like the respiratory and circulatory systems, as well as the kidneys and liver, it has considerable deficiencies in evaluating endothelial damage [26]. Complementarily, EASIX underscores the critical role of endothelial dysfunction in the treatment of AKI, enhancing our overall comprehension of its underlying pathology. It is well established that those experiencing advanced age or stage of AKI, CKD, sepsis and CRRT, and those exhibiting increased APS III are more likely to encounter adverse outcomes. However, the EASIX demonstrated a more pronounced effect on the remaining patient population, thereby illustrating its efficacy to identify the residual risks of adverse outcome events not been addressed from the previously mentioned factors. This highlights the need of

monitoring EASIX levels, even in those seeming clinically stable or exhibiting recovery signs, as heightened EASIX levels remain associated with unfavorable survival rates in the subsequent year. Furthermore, the interplay of oxidative stress, inflammation and damage to microvascular endothelium exacerbate the progression of AKI [27,28]. This study identified that albumin levels, serving as an inflammatory marker, modulate the influence of the logarithmic transformation of EASIX on mortality risk in patients with AKI, highlighting this intricate interaction. In conclusion, our findings advocate for the preferential use of the EASIX score in patients who are relatively young, exhibit a milder AKI grade, do not have CKD or sepsis, are not undergoing CRRT, or possess low APS III and high albumin levels.

EASIX may indicate systemic endothelial dysfunction, which is closely associated with inflammation and oxidative stress, and may aid in the development of therapeutic drugs. For instance, statin-based endothelial protection improves survival in allogeneic stem cell transplant patients, especially with intermediate endothelial risk (EASIX scores) [29]. Glucocorticoids are frequently employed in the management of various conditions in critically ill patients, such as sepsis and AKI. In cases of sepsis-induced AKI, glucocorticoid use, particularly with hydrocortisone, has been observed to inhibit pro-inflammatory cytokines and to enhance serum creatinine levels and urine output, indicating potential improvements in renal function [30]. Nevertheless, there is increasing concern regarding the safety of glucocorticoids in patients with AKI, especially concerning their long-term effects. While acute administration may mitigate inflammation and improve survival in critical situations, prolonged glucocorticoid use may result in metabolic adverse effects, potentially complicating patient outcomes [31]. During the COVID-19 pandemic, the use of high-dose glucocorticoids was linked to a heightened risk of severe complications in patients with pre-existing kidney disease [32]. Previous exposure to glucocorticoids may be associated with an increased mortality and a heightened risk of complications related to AKI [33]. Our analysis indicated that glucocorticoid administration may have adverse effects on this patient cohort. However, these effects appeared to be mitigated in individuals with elevated EASIX levels. The underlying mechanism may lie in the targeted reversal of the pathological state of endothelial by glucocorticoids. Studies have shown that glucocorticoids could enhance endothelial barrier function by activating the SphK1-S1P signaling pathway and inhibit the expression of adhesion molecules such as ICAM-1 by inhibiting the NF-κB pathway, thereby reducing vascular leakage and inflammatory storms [34,35]. High EASIX may indicate impaired endothelial barrier integrity and microcirculatory disturbances, and the protective effects of glucocorticoids could counteract this core pathophysiological process, thereby counteracting the potential side effects of the drug, manifested by a decrease in risk. On the contrary, patients in the low EASIX group had milder endothelial stress and a narrow window for glucocorticoid treatment, and their non-specific immunosuppressive and metabolic side effects may outweigh the benefits of endothelial protection, leading to increased risk. Based on this, the modulation of therapeutic outcomes by EASIX could facilitate the refinement and optimization of EASIX-guided treatment strategies. For instance, in patients with elevated EASIX, we may recommend against the outright rejection of corticosteroids while addressing indications for AKI or other conditions. Conversely, in patients exhibiting declining EASIX levels, we may advise a cautious approach to glucocorticoid use and encourage the exploration of alternative therapies for other indications. Although these findings necessitate validation through randomized controlled trials, the proposed analyses provide valuable insights, particularly in patients with high levels of oxidative stress, inflammation and EASIX. In conclusion, we hypothesize that it may be possible to identify or develop pharmacological agents effective for endothelial injury-related diseases based on EASIX assessments. This would strongly promote the transformation of EASIX's clinical value.

However, this research has its limitations. Firstly, although EASIX is presently viewed as a reflection of systemic endothelial dysfunction, it might not specifically evaluate the extent of renal microvascular endothelial impairment. The creation of accessible and straightforward serum markers that are specific to organs might signify a valuable avenue for future investigations. Additionally, EASIX should be examined for its correlation with imaging findings and even histopathological assessments of the kidney, to provide more pertinent high-quality evidence. Secondly, even after adjusting for major risk factors, unmeasured confounding variables may still exist due to limitations inherent in the database. Thirdly, the absence of data resulted in the exclusion of a portion of patients, which could potentially lead to population selection bias.

## Conclusions

This study indicated that higher log2(EASIX) levels were independently linked to a heightened risk of all-cause mortality in severe patients with AKI, with more pronounced effect noted in patients who were younger, had elevated albumin levels or reduced APS III, and those characterized by mild AKI, and absence of CKD, sepsis or CRRT. Additionally, this association may lead to an optimization of anti-inflammatory treatment strategies for these patients. These findings require validation through well-designed prospective studies.

## Supporting information

**S1 Fig. The variance inflation factors between log2(EASIX) and the covariates in fully multivariable analysis.** All the values of the variance inflation factors were below 5, and the existence of multicollinearity was not considered. (TIF)

**S2 Fig. Feature contribution for endpoint prediction based on the Boruta algorithm.** The vertical axis was the name of each variable, and the horizontal axis is the Z value of each variable. The plot showed the Z value of each variable during model calculation. (TIF)

**S1 Table. The association between glucocorticoid use and all-cause mortality.** (DOCX)

## Author contributions

**Conceptualization:** Kai Wang.

**Data curation:** Kai Wang.

**Investigation:** Chengshu Liang, Kai Wang.

**Methodology:** Kai Wang.

**Software:** Chengshu Liang.

**Visualization:** Chengshu Liang.

**Writing – original draft:** Chengshu Liang.

**Writing – review & editing:** Kai Wang.

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
