## [Decision Letter · Decision Letter 0]

2 Apr 2026

PONE-D-25-27246

Endothelial activation and stress index in risk stratification and treatment optimization for critically ill patients with acute kidney injury: A retrospective cohort study from MIMIC-IV database

PLOS One

Dear Dr. Wang,

Thank you for submitting your manuscript to PLOS ONE. After careful consideration, we feel that it has merit but does not fully meet PLOS ONE’s publication criteria as it currently stands. Therefore, we invite you to submit a revised version of the manuscript that addresses the points raised during the review process.

We look forward to receiving your revised manuscript.

Kind regards,

Chiara Lazzeri

Academic Editor

PLOS One

**Journal Requirements:**

Reviewers' comments:

Reviewer's Responses to Questions

**Comments to the Author**

1. Is the manuscript technically sound, and do the data support the conclusions?

Reviewer #1: Yes

Reviewer #2: Yes

2. Has the statistical analysis been performed appropriately and rigorously?

Reviewer #1: Yes

Reviewer #2: Yes

3. Have the authors made all data underlying the findings in their manuscript fully available?

Reviewer #1: Yes

Reviewer #2: Yes

4. Is the manuscript presented in an intelligible fashion and written in standard English?

Reviewer #1: Yes

Reviewer #2: Yes

5. Review Comments to the Author

Reviewer #1: This study leverages the large sample size (n=17,624) of the MIMIC-IV database to investigate, for the first time, the value of the Endothelial Activation Stress Index (EASIX) in predicting outcomes and optimizing treatment for critically ill patients with AKI. The study design is well-structured, employing rigorous statistical methods (including Cox models, Boruta algorithm, etc.), and the conclusions hold potential clinical significance. However, certain methodological details require clarification, the presentation of results needs refinement, and the discussion section warrants greater caution. Recommended for conditional acceptance (Major Revision).

1.The abstract and results claim that “glucocorticoid-associated risks were mitigated in patients with elevatedlog2(EASIX) levels” (p<0.001), but the main text only demonstrates this trend via the RCS curve in Figure 4D. This suggests that the 1-year mortality rate decreases in patients using glucocorticoids as log2(EASIX) increases. We recommend revising the description accordingly.

2.There were 26.8% of patients who received glucocorticoid therapy, which may adversely affect patient prognosis (Figure 4D). We recommend adding a subgroup analysis for glucocorticoid use, supplementing with the HR values for glucocorticoids and the interaction P-values in the high/low EASIX groups, respectively.

3.It is recommended to include a discussion and analysis of the potential mechanisms underlying the reduction in mortality associated with glucocorticoid use in high-level EASIX groups.

4.It is recommended that figures and charts in the manuscript be standardized according to journal requirements, including but not limited to units, abbreviation explanations, and numerical formats. Image quality needs improvement.

Reviewer #2: Thank you for the opportunity to review this interesting manuscript. I have read the paper with great interest. This study provides a rigorous and well-executed retrospective analysis of the endothelial activation and stress index in critically ill patients with acute kidney injury using the MIMIC-IV database. The methodology is robust, incorporating advanced statistical techniques such as overlap-weighted propensity scores, restricted cubic splines, and the Boruta algorithm. The findings are clinically relevant and offer valuable insights into risk stratification and potential treatment optimization. Overall, my impression of this manuscript is highly positive, and I believe it is a strong candidate for publication after minor revisions.

1. Please standardize the database name throughout (e.g., MIMIC-Ⅳ), avoiding mixed forms such as “MIMIC - IV,” “MIMIC - Ⅳ,” etc. The Methods state “from 2008 and 2022,” which should read “from 2008 to 2022”

2. EASIX is defined using the first measurement within 24 hours of ICU admission, but some patients may have already developed AKI prior to ICU admission. Please clarify in the Methods the exact AKI identification/ascertainment window and how this aligns temporally with the exposure definition, to reduce concerns about reverse causality.

3. The manuscript uses “relative ratio” in places, please consider standardizing to relative risk (RR) where modified Poisson regression is used.

4. Please standardize units/capitalization (e.g., mg/dL rather than mg/dl) in Table 1.

5. There is a very minor inconsistency in the spelling of "Poisson" regression in Table 2 (currently written as "Poison regression"). This is a trivial typographical error that should be corrected.

6. In Table 2, the hazard ratios (HRs) for “Each unit increase” show a pattern of initially decreasing and then increasing across different models (model1 1.14，model2 1.20，model3 1.17). I would suggest that the authors briefly explain this in the Table note to avoid confusion for readers.

7. In the figure legend the manuscript uses “shaded feature,” whereas the main text alternates between “shadow feature” and “shaded feature.” Since the standard Boruta terminology is “shadow features,” please unify the wording throughout (text, legends, and any supplementary materials) to avoid confusion.

6. PLOS authors have the option to publish the peer review history of their article (what does this mean? ). If published, this will include your full peer review and any attached files.

**Do you want your identity to be public for this peer review?**  For information about this choice, including consent withdrawal, please see our Privacy Policy .

Reviewer #1: No

Reviewer #2: No

---

## [Author Response · Author response to Decision Letter 1]

10 Apr 2026

Dear Editor and Reviewers,

Many thanks to you for your valuable advice and recognition of our manuscript titled "Endothelial activation and stress index in risk stratification and treatment optimization for critically ill patients with acute kidney injury: A retrospective cohort study from MIMIC-IV database" (Submission ID: PONE-D-25-27246). We sincerely appreciate the time and expertise you have invested in reviewing our work, as your insightful comments have provided not only essential guidance for revising the paper but also important direction for our research. We have carefully studied each comment and made targeted corrections that we believe address your concerns. Specific locations of modifications are highlighted in Red in the revised manuscript, and the comments and suggestions point by point are detailed in the attached Response document.

We look forward to your positive response to the revised work submitted here. PLoS One is an influential journal which aims to improve our understanding of diseases. From the papers published in your journal, we have been learning a lot. Hopefully, we could have our article been considered of publication in your journal. Should there been any other corrections we could make, please feel free to contact us.

Sincerely,

Kai Wang

Department of Cardiology, The Second Affiliated Hospital of Chongqing Medical University, Chongqing, China.

Email: nkuwangkai@163.com.

Editor's comments

Comment 1: Please include the following items when submitting your revised manuscript:

A letter that responds to each point raised by the academic editor and reviewer(s). You should upload this letter as a separate file labeled 'Response to Reviewers'.

Response: Sincerely, thank you again for your efforts and contributions to our manuscript. Following your instructions:

(1) We responded positively to every point and comment and uploaded it as a separate file labeled 'Response to Reviewers'.

(2) We created a new copy of the manuscript and highlighted all the revisions in red font. This copy is uploaded as a separate file labeled 'Revised Manuscript with Track Changes'.

(3) We upload the revised manuscript as a separate file labeled 'Manuscript'. The manuscript does not contain any traces or markings.

Comment 2: When submitting your revision, we need you to address these additional requirements.

Your ethics statement should only appear in the Methods section of your manuscript. If your ethics statement is written in any section besides the Methods, please delete it from any other section.

Response: Thank you for your detailed guidance.

(1) We have carefully read both the two documents and the recent publications of PLoS One. Based on this information, we adjusted the manuscript's style requirements as appropriate.

(2) We have removed the section of "Ethical approval" between the section of "Author contributions" and section of "Disclosure of interest". i.e.

Ethical approval

Approval for the study was granted by the Human Institutional Review Board at Beth Israel Deaconess Medical Center. The studies were conducted in accordance with the local legislation and institutional requirements. The further ethical review was considered unnecessary for this study, as no additional data collection was undertaken. The individual patient consent was deemed unnecessary for this study, as the study was retrospective and the anonymized data was publicly available, and individuals could not be identified either directly or indirectly.

(3) The comments of the two reviewers did not include suggestions to cite specific previously published works. Furthermore, under the professional guidance of editor and reviewers, the quality of manuscripts has been improved dramatically. We believe that the content of the manuscript would attract the attention of peer researchers and our team will actively cite this manuscript as a basis for further research.

Response to Reviewer 1

Comment 1: This study leverages the large sample size (n=17,624) of the MIMIC-IV database to investigate, for the first time, the value of the Endothelial Activation Stress Index (EASIX) in predicting outcomes and optimizing treatment for critically ill patients with AKI. The study design is well-structured, employing rigorous statistical methods (including Cox models, Boruta algorithm, etc.), and the conclusions hold potential clinical significance. However, certain methodological details require clarification, the presentation of results needs refinement, and the discussion section warrants greater caution. Recommended for conditional acceptance (Major Revision).

Response: We are very grateful for your encouraging feedback and recognition of our work, which is a great honor for our team. Under the guidance of you and other experts, We actively clarified some methodological details, improved the presentation of results, and revised some of the discussion.

Comment 2: The abstract and results claim that "glucocorticoid-associated risks were mitigated in patients with elevatedlog2(EASIX) levels" (p<0.001), but the main text only demonstrates this trend via the RCS curve in Figure 4D. This suggests that the 1-year mortality rate decreases in patients using glucocorticoids as log2(EASIX) increases. We recommend revising the description accordingly.

Response: Thank you for your detailed guidance, which indeed highlights an important aspect. Taking into account your comment 3, we have supplemented the HR value of glucocorticoid therapy. Based on this, we modified the description of the glucocorticoid-related risk and the risk change with log2 (EASIX) level change in the Abstract section and the Results section. Line 30-32 and 206-210 in the Revised Manuscript with Track Changes. i.e.

"Abstract

Results: ... The glucocorticoid use may be independently associated with an increased risk of 1-year (HR: 1.27, 95% CI: 1.21-1.34) and ICU (HR: 1.39, 95% CI: 1.31-1.47) mortality. The glucocorticoid-associated risk decreased as the log2(EASIX) level increased (p < 0.001)."

"Results

Subgroup analysis

...

About a quarter of patients (26.8%) of patients underwent glucocorticoid therapy. After minimizing confounding bias, this glucocorticoid use was independently associated with an increased risk of 1-year (HR: 1.27, 95% CI: 1.21-1.34) and ICU (HR: 1.39, 95% CI: 1.31-1.47) mortality (S1 Table). Meanwhile, the glucocorticoid-associated risk decreased as the log2(EASIX) level increased (p for interaction < 0.001) (Fig 4D, S1 Table)."

Comment 3: There were 26.8% of patients who received glucocorticoid therapy, which may adversely affect patient prognosis (Figure 4D). We recommend adding a subgroup analysis for glucocorticoid use, supplementing with the HR values for glucocorticoids and the interaction P-values in the high/low EASIX groups, respectively.

Response: Thank you for your insightful guidance. Following your instructions, We have added subgroup analyses of glucocorticoid use and corresponding interaction p-value, which are presented in the forest plot. We have supplemented the HR value of glucocorticoid use and the interaction p-value of the high/low EASIX groups, which are presented in Supplementary table 1. In addition, we have revised the relevant description to take into account your comment 2. Line 197-199 and 206-210 in the Revised Manuscript with Track Changes. i.e.

"Results

Subgroup analysis

...

Conversely, the link was notably stronger in those experiencing mild AKI (p for interaction = 0.007), those who did not have CKD (p = 0.047), CRRT (p = 0.016), sepsis (p < 0.001) or glucocorticoid use (p < 0.001).

...

About a quarter of patients (26.8%) of patients underwent glucocorticoid therapy. After minimizing confounding bias, this glucocorticoid use was independently associated with an increased risk of 1-year (HR: 1.27, 95% CI: 1.21-1.34) and ICU (HR: 1.39, 95% CI: 1.31-1.47) mortality (S1 Table). Meanwhile, the glucocorticoid-associated risk decreased as the log2(EASIX) level increased (p for interaction < 0.001) (Fig 4D, S1 Table)."

"

Fig 3. The restricted cubic spline curve for all patients and forest plot for subgroup analysis. (A) Restricted cubic spline curve of the log2(EASIX) and hazard ratios in patients with AKI. (B) Subgroup analysis by gender, diabetes, hypertension, AKI stage, heart failure, sepsis, chronic kidney disease, glucocorticoid use, continuous renal replacement therapy, and ventilation."

"S1 Table. The association between glucocorticoid use and all-cause mortality.

glucocorticoid number model 0 model 1 model 2 model 3 p for interaction

17624 Hazard Ratio for 1-year mortality (Cox regression) -

No 12895 reference reference reference reference

Yes 4729 1.54 (1.47, 1.61) 1.47 (1.40, 1.54) 1.29 (1.23, 1.36) 1.27 (1.21, 1.34)

11346 Only in patients with low-level log2(EASIX) < 0.001

No 8635 reference reference reference reference

Yes 2711 1.57 (1.47, 1.67) 1.54 (1.45, 1.65) 1.42 (1.33, 1.52) 1.42 (1.32, 1.51)

6278 Only in patients with high-level log2(EASIX)

No 4260 reference reference reference reference

Yes 2018 1.34 (1.25, 1.44) 1.32 (1.23, 1.42) 1.12 (1.05, 1.21) 1.10 (1.02, 1.18)

17624 Relative Risk for ICU mortality (modified Poisson regression) -

No 12895 reference reference reference reference

Yes 4729 1.88 (1.77, 1.99) 1.65 (1.56, 1.75) 1.42 (1.34, 1.51) 1.39 (1.31, 1.47)

11346 Only in patients with low-level log2(EASIX) < 0.001

No 8635 reference reference reference reference

Yes 2711 1.92 (1.75, 2.11) 1.73 (1.58, 1.90) 1.50 (1.37, 1.65) 1.48 (1.34, 1.62)

6278 Only in patients with high-level log2(EASIX)

No 4260 reference reference reference reference

Yes 2018 1.62 (1.50, 1.74) 1.50 (1.39, 1.61) 1.30 (1.21, 1.40) 1.28 (1.18, 1.37)

model 0: only glucocorticoid use was included.

model 1: age, gender, diabetes, hypertension, AKI stage, CKD, heart failure, cirrhosis, sepsis, and cancer were adjusted.

model 2: age, gender, diabetes, hypertension, AKI stage, CKD, heart failure, cirrhosis, sepsis, and cancer, as well as WBC, hemoglobin, RDW, platelets, AST, ALT, TBil, albumin, creatinine, BUN, LDH, lactate, anion gap, antibiotic, RAASi, ventilation, and CRRT were adjusted.

model 3: age, gender, diabetes, hypertension, AKI stage, CKD, heart failure, cirrhosis, sepsis, cancer, WBC, hemoglobin, RDW, platelets, AST, ALT, TBil, albumin, creatinine, BUN, LDH, lactate, anion gap, antibiotic, RAASi, ventilation, and CRRT, as well as SOFA and APS Ⅲ were further adjusted."

Comment 4: It is recommended to include a discussion and analysis of the potential mechanisms underlying the reduction in mortality associated with glucocorticoid use in high-level EASIX groups.

Response: We thank the reviewer for this important consideration. In the fourth paragraph of the Discussion section, we have discussed the associated content of the reduction in mortality associated with glucocorticoid use in the high-level EASIX group. However, it did not include the underlying mechanism. Following your instructions, we have revised some expressions and expanded the Discussion in this section to include an analysis of potential mechanisms. Line 284-307 in the Revised Manuscript with Track Changes. i.e.

"Discussion

...

In cases of sepsis-induced AKI, glucocorticoid use, particularly with hydrocortisone, has been observed to inhibit pro-inflammatory cytokines and to enhance serum creatinine levels and urine output, indicating potential improvements in renal function[32]. Nevertheless, there is increasing concern regarding the safety of glucocorticoids in patients with AKI, especially concerning their long-term effects. While acute administration may mitigate inflammation and improve survival in critical situations, prolonged glucocorticoid use may result in metabolic adverse effects, potentially complicating patient outcomes[33]. During the COVID-19 pandemic, the use of high-dose glucocorticoids was linked to a heightened risk of severe complications in patients with pre-existing kidney disease[34]. Previous exposure to glucocorticoids may be associated with an increased mortality and a heightened risk of complications related to AKI[35]. Our analysis indicated that glucocorticoid administration may have adverse effects on this patient cohort. However, these effects appeared to be mitigated in individuals with elevated EASIX levels. The underlying mechanism may lie in the targeted reversal of the pathological state of endothelial by glucocorticoids. Studies have shown that glucocorticoids could enhance endothelial barrier function by activating the SphK1-S1P signaling pathway and inhibit the expression of adhesion molecules such as ICAM-1 by inhibiting the NF-κB pathway, thereby reducing vascular leakage and inflammatory storms[36,37]. High EASIX may indicate impaired endothelial barrier integrity and microcirculatory disturbances, and the protective effects of glucocorticoids could counteract this core pathophysiological process, thereby counteracting the potential side effects of the drug, manifested by a decrease in risk. On the contrary, patients in the low EASIX group had milder endothelial stress and a narrow window for glucocorticoid treatment, and their non-specific immunosuppressive and metabolic side effects may outweigh the benefits of endothelial protection, leading to increased risk. Based on this, the modulation of therapeutic outcomes by EASIX could facilitate the refinement and optimization of EASIX-guided treatment strategies. ..."

Comment 5: It is recommended that figures and charts in the manuscript be standardized according to journal requirements, including but not limited to units, abbreviation explanations, and numerical formats. Image quality needs improvement.

Response: Thanks for the meticulous advice. Following your and the editor's guidance, we have standardized the format and content of the manuscript in accordance with the journal's requirements. In addition, our Images are all 300dpi resolution. In the submission system - PDF file, the image is compressed. In the PDF file used for peer review, a download link to the original image is provided in the upper right corner of the Figure page. Again, we would like to express our sincere gratitude for your valuable efforts and professional evaluation.

Response to Reviewer 2

Comment 1: I have read the paper with great interest. This study provides a rigorous and well-executed retrospective analysis of the endothelial activation and stress index in critically ill patients with acute kidney injury using the MIMIC-IV database. The methodology is robust, incorporating advanced statistical techniques such as overlap-weighted propensity scores, restricted cubic splines, and the Boruta algorithm. The findings are clinically relevant and offer valuable insights into risk stratification and potential treatment optimization. Overall, my impression of this manuscript is highly positive, and I believe it is a strong candidate for publication after minor revisions.

Response: Thank you very much for your comprehensive evaluation of our manuscript and for your encouraging comments on our team's work. Under the guidance of you experts, we actively completed the revision of the manuscript. Our team has always been committed to scientific research close to the clinic and hopes to continue to maintain this academic

---

## [Editor Report · Decision Letter 1]

20 Apr 2026

Endothelial activation and stress index in risk stratification and treatment optimization for critically ill patients with acute kidney injury: A retrospective cohort study from MIMIC-IV database

PONE-D-25-27246R1

Dear Dr. Wang,

We’re pleased to inform you that your manuscript has been judged scientifically suitable for publication and will be formally accepted for publication once it meets all outstanding technical requirements.

Kind regards,

Chiara Lazzeri

Academic Editor

PLOS One
---

## [Editor Report · Acceptance letter]

PONE-D-25-27246R1

PLOS One

Dear Dr. Wang,

I'm pleased to inform you that your manuscript has been deemed suitable for publication in PLOS One. Congratulations! Your manuscript is now being handed over to our production team.

Kind regards,

on behalf of

Dr. Chiara Lazzeri

Academic Editor

PLOS One